# A Strategy Based on GC-MS/MS, UPLC-MS/MS and Virtual Molecular Docking for Analysis and Prediction of Bioactive Compounds in *Eucalyptus Globulus* Leaves

**DOI:** 10.3390/ijms20163875

**Published:** 2019-08-08

**Authors:** Meng Pan, Qicheng Lei, Ning Zang, Hong Zhang

**Affiliations:** 1School of Food Science and Biological Engineering, Zhejiang Gongshang University, Hangzhou, Zhejiang 310035, China; 2Guangxi Medical Research Center, Guangxi Medical University, Nanning, Guangxi 530021, China

**Keywords:** *Eucalyptus globulus*, GC-MS/MS, UPLC-MS/MS, molecular docking, allelopathic test

## Abstract

The discovery of medicinal plants is crucial for drug development. *Eucalyptus globulus* leaves are used as a traditional medicine in many areas of world due to herbicidal and insecticidal activity. While natural products are difficult to be separated and activity assayed, a new approach is needed to predict the active ingredients therein. In this study, a new method for screening active compounds extracted from *E. globulus* leaves was developed by GC-MS/MS and UPLC-MS/MS combined with molecular docking technology. Predicted compounds with high activity were proposed. Firstly, 35 volatile compounds and 34 aqueous extracted compounds were extracted from *E. globulus* leaves, and identified by GC-MS/MS and UPLC-MS/MS. The herbicidal receptor (1BX9) was then docked with the identified compounds by docking software, evaluated by docking models and seven scoring functions. The results showed that gallic acid had a strong inhibitory activity of 1BX9, which was speculated to be the main reason for the inhibitory effect of *E. globulus* leaves. Finally, allelopathic tests of gallic acid, citric acid, and isopulegol were carried out on grass seeds to verify its inhibitory activity against herbicide receptor 1BX9. The results show that the method can screen compounds with specific activity from a complex system of medicinal plants, which is very important for the screening of new active ingredients, confirmation of new medicinal ingredients, and the in-depth development of animal and plant medicines.

## 1. Introduction

*Eucalyptus globulus* Labill, an evergreen plant, is known in China as a fast-growing genus. It is a myrtaceae plant, usually distributed in tropical and subtropical regions, and also in the Guangdong, Guangxi and Yunnan provinces in China [1]. It has both pharmacological and biotoxicity activities. Pharmacological activities mainly include antioxidant, anti-tumor, hypoglycemia, etc., and biotoxicity activities include allelopathy, insecticidal action and antimicrobial activity [2,3].

At present, the research on the allelopathic activity of *E. globulus* leaves is mainly on the essential oil, but the allelopathic effect on the aqueous extracts of *E. globulus* leaves is still less known. There are four ways to release the allelopathic active ingredients: volatile release, root release, leaching release and decay release [4]. The leaching of *E. globulus* leaves is a way to release the active compounds of *E. globulus* leaves. It spreads through water and affects the environment. Therefore, the aqueous extracts of *E. globulus* leaves may contain a variety of important biologically active compounds. Therefore, we study the allelopathic activity of the volatile components and aqueous extracted components of *E. globulus* leaves.

Traditional separation techniques for natural products have low separation efficiency and high cost. More importantly, the active ingredients are usually low in content, and disappear during the separation process due to the irreversible adsorption of solid supports. Sometimes the recovery of the isolated compound is too low to allow further activity testing [5,6]. Over the last decade, GC-MS/MS, UPLC-MS/MS have been widely used in separation and rapid identification of compounds in natural products [7]. However, only using chromatographic-mass spectrometry, it is not possible to determine which compound in the complex mixture responds to which activity. To cope with this problem, the molecular docking method [8] was introduced.

The molecular docking technique uses energy-based scoring functions to predict interactions between ligand and receptor active sites [9]. It simulates the interaction between small molecule ligands and biomacromolecular receptors based on the "lock-key" principle. Ligand-receptor interaction is a process of molecular recognition, which mainly includes electrostatic interaction, hydrogen bonding, hydrophobic interaction, van der Waals interaction, etc. Through calculation, the combination mode and affinity between them can be predicted to perform virtual screening of drugs.

In this study, the compounds of *E. globulus* leaves were predicted by GC-MS/MS and LC-MS/MS combined with molecular docking technology. The potential active ingredients were determined by scoring function, interaction, etc. Finally, the predicted activity of active ingredients was verified by an activity test, and the active compounds of the *E. globulus* leaves were screened.

## 2. Results and Discussion

### 2.1. Condition Optimization

#### 2.1.1. Optimization of UPLC Conditions

In order to obtain the best separation effect of the components in the aqueous extracts from the *E. globulus* leaves, the separation conditions of the ultra-high-performance liquid chromatography (UPLC) were optimized in this study, including the optimization of the column, mobile phase and column temperature.

Columns of different lengths and particle sizes were tested. The optimization of mobile phase conditions included the separation effect of methanol-formic acid solution and acetonitrile-formic acid solution, different mobile phase flow rates and the addition of different formic acid concentrations (0.1%, 0.5%, 1%). For optimizing the column temperature, we respectively tested 25, 30, 35, 40 °C. After the test, using the Waters Xselet HSS T3 (150 mm × 2.1 mm, 3.5 mm), acetonitrile-0.1%formic acid solution of mobile phase, flow rate of 0.2 mL/min, 30 °C column temperature, can get better separation results.

#### 2.1.2. Optimization of HS-SPME Extraction Method

The equilibrium time, extraction time and extraction temperature of the headspace solid phase microextraction (HS-SPME) method have a great influence on the volatile components in the *E. globulus* leaves. The equilibrium time is designed in this experiment (20, 30, 40 min), extraction time (30, 40, 50 min) and extraction temperature (40, 50, 60 °C), to investigate the optimal equilibrium time, extraction time and extraction temperature.

After comparing the number of volatile components and the peak intensity in GC-MS in different equilibrium time, extraction time and extraction temperature. The equilibrium time of 30 min, extraction time of 40 min and the extraction temperature of 50 °C is the optimum extraction condition.

### 2.2. Identification of Active Compounds of E. Globulus Leaves

#### 2.2.1. Aqueous Extracts from *E. Globulus* Leaves

In order to analyze the bioactive compounds of aqueous extracts from the *E. globulus* leaves, UPLC-Q-Orbitrap-MS was used, and the mass spectra of the aqueous extracts were shown in Figure 1. A total of 34 active compounds were identified, mainly including gallic acid, protocatechuic acid, quinic acid, caffeic acid, P-coumalic acid, benzoic acid and ferulic acid. Among all the compounds, the content of tannins components was the highest class, accounting for 3.45% of aqueous extracts. Among them, the content of gallic acid and ellagic acid were higher, and the content of gallic acid accounted for 2.81%, and the ellagic acid content was 0.19%. In addition, the content of phenolic acid in the aqueous extracts was the second class, reaching 2.93%, of which the quinic acid content was the highest, accounting for 2.40%, and caffeic acid, protocatechuic acid and gentisic acid both accounted for 0.15% (Table 1). It can be found that gallic acid is the most abundant component in the aqueous extracts. Gallic acid is one of the common ingredients in eucalyptus plants. There are many studies on the antibacterial activity of gallic acid, which has a good inhibitory effect on Staphylococcus aureus and Salmonella enteritidis [10]. It is recognized in Eucalyptus robusta smith and Eucalyptus urophylla [11] and may be highly correlated with the activity of the aqueous extracts from *E. globulus* leaves.

#### 2.2.2. Volatile Compounds of *E. Globulus* Leaves

The mass spectra of volatile components of *E. globulus* leaves were obtained by HS-SPME-GC-MS, as shown in Figure 2. A total of 35 volatile compounds were identified, mainly including isopulegol, α-terpineol, β-eudesmol, γ-terpinene, 3-carene, β-pinene, α-pinene, and camphene. Each volatile component of the *E. globulus* leaves was well separated and identified. The monoterpenoid volatile component was abundant, accounting for 43.09% of the total volatile content, which was the most important volatile component. Studies have found that the various biological activities are closely related to the monoterpenoids, and the monoterpenoids extracted from many plants have biological activity or potential biological activity. Isopulegol belongs to the monoterpenoids, accounting for 0.41% of the volatile compounds from *E. globulus* leaves (Table 2). It presents in the essential oils of many plants, such as Corymbia citriodora H. [21], Zanthoxylum schinifolium L [22]. and Melissa officinalis L. [23], all these plants present anti-inflammatory activities. Isopulegol also has central and peripheral analgesic effects [24], as well as a broad spectrum of antifungal activity [25]. It may be related to the biological activity of the volatile compounds in *E. globulus* leaves.

### 2.3. Prediction of Active Components of E. Globulus Leaves

#### 2.3.1. Docking Results

Using herbicide as a key word, the relevant protein was searched from the TTD database as a target protein for herbicide, and the corresponding literature was searched and screened. The crystal structure can be obtained from the PDB database (http://www.rcsb.org/pdb/home/home.do), and the herbicidal protein is glutathione s-transferase and herbicide complex (1BX9). The Ligandfit [26] package in Discovery Studio is a classic tool for virtual screening of molecular docking methods. It has the functions of automatic searching and confirmation of receptor active sites, multi-ligand docking of conformational flexibility, and evaluation of interaction scores based on force field.

After docking 69 active compounds (Appendix A) from *E. globulus* leaves with the target protein, it was found that 23 aqueous extracted compounds and 29 volatile compounds were successfully docked with 1BX9. As a result, the gallic acid had a maximum score of 89.142 and also the highest content in the aqueous extracted compounds, so it was suspected to have potential 1BX9 inhibitory activity. The dock score of glucogallic acid was also high at 82.628. Glucogallic acid is the glycosylation of gallic acid, which has enhanced aqueous solubility and low toxicity [27]. The scoring value of it was lower than gallic acid, which conforms to the docking result (Table 3). The citric acid dock score in the aqueous extracts was ranked third at 76.598, had potential 1BX9 inhibitory activity. Among the volatile compounds from the *E. globulus* leaves, the isopulegol dock score had the highest scoring value and belonged to the monoterpenoids, which may have strong biological activity. It can be seen from Table 3 that most aqueous extracted compounds from *E. globulus* leaves were superior to the volatile compounds in the docking result of 1BX9.

#### 2.3.2. Docking Model

Among all identified compounds from *E. globulus* leaves, gallic acid, glucogallic acid, and citric acid scored higher on the dock score, and all belonged to the aqueous extracts; isopulegol scored the highest in the volatile compounds. Therefore, the docking model of gallic acid, glucogallic acid, citric acid, and isopulegol were analyzed (Table 4) to study the interaction of the active compounds with the herbicidal receptor (1BX9).

The greater the absolute energy in the docking model, the better the docking result, and the dashed line indicates the hydrogen bond formed with the amino acid side chain. The H-Bind number indicates the number of hydrogen bond formation, and the surrounding residues also participate in the interaction. As a result, the interaction between the compound gallic acid, glucogallic acid, citric acid and 1BX9 was mainly mediated by the amino acid residue Lys 35, and the interaction between isopulegol and 1BX9 was mainly caused by the amino acid residues Lys 35 and Tyr 126 (Table 4). Both conduct and interact through the formation of hydrogen bonds. The absolute energies of gallic acid, glucogallic acid, citric acid and isopulegol were 15.277, 13.502, 12.893 and 4.431 kcal/mol, respectively. Among them, gallic acid had the highest absolute energy, indicating that gallic acid had a good docking result with 1BX9. In addition to these residues, amino acid residues (His 7, Ile 125, Ala 9, Pro 8) also promoted substrate recognition by combination of electrostatic, hydrophobic interactions. The scoring and docking model analysis showed that gallic acid had the strongest potential inhibitory activity.

### 2.4. Activity Verification

#### 2.4.1. Allelopathic Effect of *E. Globulus* Leaves Extract

In order to study the allelopathic effect of the extract from *E. globulus* leaves on grass seeds, distilled water was used as a control, the aqueous extracted compounds and volatile compounds were diluted to different concentrations. The germination rate, root inhibition rate and seedling inhibition rate of grass seeds were observed to evaluate their allelopathic effects. In this experiment, the lower germination rate, the better germination effect of inhibiting grass seeds. However, the inhibition rate of roots and seedlings is opposite, the higher inhibition rate, the better allelopathic effect. The results showed that aqueous extracted compounds and volatile compounds all had an allelopathic effect, and the concentration of the extract had a positive correlation with the allelopathic effect of the grass seeds. As the concentration increases, the germination rate of grass seeds gradually decreased, while the inhibition rate of root growth and seedling growth gradually increased. At the same concentration, the germination rate of the aqueous extracted compounds was lower than that of the volatile compounds on the grass seed. The inhibition rate of the grass seedlings and roots was higher than the volatile compounds. It indicated that the allelopathic effect of aqueous extracted compounds on grass seeds were higher than volatile compounds from *E. globulus* leaves.

#### 2.4.2. Allelopathic Effects of Predicted Active Ingredients of *E. Globulus* Leaves

The above allelopathic experiments showed that the allelopathic effect of the aqueous extracted compounds were stronger than the volatile compounds from *E. globulus* leaves, which was consistent with the predicted results of molecular docking. Molecular docking technology predicted that the most allelopathic effect in E. globulus leaves was gallic acid, followed by gallic acid glucoside and citric acid. Because gallic acid glucoside is the glycosylation of gallic acid, its aqueous solubility is enhanced and its toxicity is reduced [27], therefore the allelopathic effect on grass seeds is lower than that of gallic acid. In the volatile component, isopulegol was predicted to have the strongest allelopathic effect, so the allelopathic effect of gallic acid, citric acid, and isopulegol was further verified.

Taking distilled water as a control, gallic acid, citric acid and isopulegol were diluted to different concentrations, through germination rate, inhibition rate of root growth and seedling growth to research their allelopathic effects on grass seeds. The results showed that gallic acid, citric acid and isopulegol had significantly inhibited the germination, seedling and root growth of grass seeds compared with the control group, so they all had allelopathic effects. The concentration was positively correlated with the allelopathic effect on grass seeds. With the increase of concentration, the germination rate of grass seeds gradually decreased, while the root growth inhibition rate and seedling growth inhibition rate gradually increased. At the same concentration, the gallic acid inhibition rates of germination and grass seedling and root on grass seeds were higher than that of citric acid and isopulegol. Comparing Figure 3 and Figure 4, it can be found that the gallic acid inhibition rates of germination, seedling and root on grass seeds were higher than that of aqueous extracted compounds, which was much higher than volatile compounds. The citric acid inhibition rates of germination and grass seedling and root on grass seeds were slightly lower than that of gallic acid, higher than that of aqueous extracted compounds and volatile compounds from *E. globulus* leaves. Among them, isopulegol had the weakest allelopathic effect, which had the lowest inhibition rate on grass seed germination, root and seedling.

Through the above experiments, it can be concluded that the aqueous extracted compounds and volatile compounds from *E. globulus* leaves, as well as the active compounds predicted by molecular docking have a certain allelopathic effect. The allelopathic effect of the aqueous extracted compounds were higher than the volatile components, and the allelopathic effects of different components were: gallic acid > citric acid > isopulegol. Gallic acid had the strongest allelopathic effect, and the allelopathic effect of isopulegol was the weakest, consistent with the results of molecular docking. In the result of molecular docking, the aqueous extracted compounds are superior to the volatile compounds, and the gallic acid was better than citric acid and isopulegol.

## 3. Experimental

### 3.1. Chemicals and Plant Materials

*E. globulus* leaves (fresh and non-destructive); grass seeds (Cynodon dactylon, purchased from garden flower seed shop, Hangzhou, China); gallic acid (purchased from source leaf standard material center, Shanghai, China); isopulegol (purchased from Tan ink Standard Substance Center, Beijing, China); citric acid (purchased from Tan ink Standard Substance Center, Beijing, China); Acetophenone, n-tetradecane (purchased from Aladdin, Los Angeles, CA, USA)

### 3.2. Extraction of Leaves

#### 3.2.1. Extraction of Aqueous Extracts from *E. Globulus* Leaves

Fresh *E. globulus* leaves (10 g) were randomly weighed, 100mL of pure aqueous at a ratio of 1:10 (*M*/*V*) was added, stirred and mixed, ultrasonically extracted for 30 min, leaching for 24 h at room temperature, and then centrifuged for 10 min, the supernatant was the aqueous extracts of *E. globulus* leaves.

#### 3.2.2. Extraction of Volatile Components from *E. Globulus* Leaves

Fresh *E. globulus* leaves (100 g) were randomly weighed and placed in a Q-250A3 pulverizer for pulverization, and then passed through a 100-mesh sieve. 1.00g (±1%) pulverized *E. globulus* leaves were weighed into a 20 mL headspace vial for testing, and the above tests were repeated twice.

### 3.3. GC-MS/MS, UPLC-MS/MS Analysis

#### 3.3.1. UPLC-MS/MS Analysis of Aqueous Extracts from Leaves

In this experiment, the active organic matter of the *E. globulus* leaves was separated by Dionex Ultimate 3000 ultra-high-performance liquid chromatography with Xselet HSS T3 (150 mm ×2.1 mm, 3.5 mm) column. The column temperature was set to 30 °C and sample tray temperature was set to 5 °C. The autosampler needle was washed twice with 200 mL of 70% methanol solution before and after each injection, detected by Waters 2996 PDA. Liquid chromatography mobile phase: phase A is acetonitrile, phase B is 0.1% formic acid, and the elution gradient is initial mobile phase 3% A for 5 min; 3–5% A, 5–10 min; 5% A for 10 min; 5–10% A, 20–25 min; 10% A for 10 min; 10–20% A, 35–50 min; 20% A for 10 min; 20–30% A, 60–70 min; 30–95% A, 70–75 min 95–3% A, 75–80 min; 3% A for 10 min. The flow rate was 0.2 mL/min and the injection volume was 10 mL.

The experiment used UPLC-Q-Orbitrap-MS, with ESI ionization source, and negative ion mode was used for scanning with the MS1 resolution setted to RP = 70,000 at 200 *m*/*z*; MS2 resolution set to RP = 35,000 at 200 *m*/*z*. The electric probe evaporator temperature was 250 °C, and scan mass range was set to 100–1000 *m*/*z*. The switching between MS1 and MS2 was performed according to the ionic strength of the substance MS1, when the ionic strength of the substance MS1 was the highest, MS2 scanning performed.

Data analysis was performed in the Xcalibur working software, and identification of some of the components was identified using the mzCloud (www.mzCloud.org) mass spectral library and NIST08f. In the identification process, the suspected parent ion is first extracted, so that a narrow mass range can be quickly determined in the complex components of each sample, and then the precise retention time, molecular ion peak and MS2 under specific energy are compared. The feature fragments are qualitatively analyzed with reference to the reference literature. Finally, the relative content was quantified by the area normalization method.

#### 3.3.2. GC-MS/MS Analysis of Volatile Components of Leaves

The main part of the headspace solid phase microextraction device is the extraction head, which needs to be aged for 1 h at 280 °C in the gas chromatographic inlet before use. The micro-extraction step was as follows: firstly add 5 mL of double internal standard (2.5 mg/mL acetophenone and 1.0 mg/mL n-tetradecane in acetone) to the headspace sample bottle containing the sample, and the headspace vial was placed in 50 °C dry bath and equilibrated for 30 min; then a 1 cm 50/30 μm DVB/CAR/PDMS extraction head was inserted into the injection vial to enrich the volatile components of the vial headspace for 40 min. The extraction head was removed and immediately inserted into the GC inlet for 5 min.

The qualitative and quantitative analysis of the volatile constituents was performed on Agilent 7890A-7000 gas chromatography with HP-5 MS capillary column (30 mm × 0.25 mm × 0.25 μm) used for separation and analysis. The injector port was set at 280 °C, helium was used as the carrier gas at a flow rate of 0.8 mL/min. The following temperature program was applied: 50 °C for 1 min, and then to 90 °C at a rate of 8 °C/min, to 130 °C at a rate of 4 °C/min, for 8 min, to 160 °C at a rate of 4 °C/min for 2 min, finally to 230 °C at a rate of 10 °C/min. Mass spectrometry uses electron bombardment ionization source (EI source), ion source temperature of 230 °C, and scan mass range of 35–400 amu.

Data analysis was performed in Agilant Mass Hunter software, and the spectra of all peaks in the gas chromatogram were compared with the NIST 2.0 database to initially characterize the peaks in the spectrum. The retention indices (RI) of the compounds were determined relative to the retention times of a series of n-alkanes (C6–C26), and then compare with the retention index of each substance under the same or similar polarity column, verify all the substances again. The internal standard method was used to quantify the relative content. The internal standard was acetophenone and n-tetradecane, and the peak area of the internal standard was compared with the peak area of the relative quantitative substance.

### 3.4. Molecular Docking

#### 3.4.1. Preparation of ligands and receptors

The *E. globulus* leaves components were collected from the TCMSP database (http://lsp.nwu.edu.cn/tcmsp.php). This database collected 499 Chinese herbal medicines registered in all Chinese Pharmacopoeias (2010) for a total of 12,144 compounds [28]. The components of *E. globulus* leaves were obtained by GC-MS/MS and UPLC-MS/MS. According to the content and activity of each component, 69 small molecules of volatile and aqueous extracts were finally screened. The three-dimensional structure of 69 small molecules was downloaded from TCMSP, imported into the docking software DiscoveryStudio2.5, hydrogenated, optimized with the CHARMm force field, and then stored as a candidate small molecule for molecular docking.

Herbicide targets are collected from the therapeutic target database (TTD, http://bidd.nus.edu.sg/group/cjttd/). The therapeutic target protein database is a specialized target database developed by the National University of Singapore. It provides information on known therapeutic proteins and nucleic acid targets, disease information and signaling pathways described in published literature [26] and plays an important role in the innovative research and development of traditional Chinese medicine. Using herbicide as keywords, the related protein was searched from the TTD database as target protein for herbicide, and the corresponding literature was searched and screened. Its crystal structure is available through the PDB database (http://www.rcsb.org/pdb/home/home.do). These targets were introduced into Discovery Studio, aqueous molecules were removed, hydrogenated, energy optimized through the CHARMm force field, and then stored as receptors.

#### 3.4.2. Docking Method

The active constituents of *E. globulus* leaves and the target protein for herbicide were docked through the Ligandfit module [29] of Discovery Studio 2.5 software (Chuangteng Technology Corporation, Beijing, China), and scored according to the energy of the system. The scoring function includes the default Ligscore1, Ligscore2 [30], PLP1, PLP2 [31], Jain [32], PMF and Dock score [29] seven scoring functions [33], other parameters are default values.

### 3.5. Allelopathic Effects on Seeds

Using the petri-dish filter method [34,35]; grass seeds (Cynodon dactylon) of the same size and full grain were selected, soaked for one day, and the cut filter paper was put in the culture dish on the second day, first wet with distilled aqueous, then 50 grass seeds were placed in each dish. The samples were then placed at different concentrations into the culture dish, 10 mL of extract was added to each dish, and three replicates were set for each concentration gradient to make the test more accurate. The control group was added with distilled aqueous. The number of seeds germinated in each petri-dish was counted from third day onwards. The final germination number was counted on the 9th day, and the root length and seedling length of all the seeds in the petri-dish were determined.

Determination of allelopathic test indicators: according to the tree seed test method GB2772-99 (Appendix A), the germination rate of seeds and the inhibition rate of root and seedling growth were measured. Grass seed growth inhibition rate = [control group root (seedling) length-experimental group root (seedling) length] / [control group root (seedling) length] × 100%. Experimental data was analyzed using EXCEL (Microsoft Excel 2016, Microsoft, Redmond, WA, USA), or Origin 8.1 software (OriginLab, Northampton, MA, USA).

## 4. Conclusions

A simple, rapid and efficient method based on GC–MS/MS, UPLC-MS/MS combined molecular docking was established for the identification of bioactive compounds in natural products. The chemical composition was first identified by mass spectrometry, the active compound was predicted by molecular docking, and the activity of the compound was verified. In this study, 69 components of *E. globulus* leaves were docked with herbicidal receptor (1BX9) to obtain the strong bioactivity of gallic acid. The activity experiment showed that gallic acid had herbicidal activity, and fully proved the effectiveness of the method, which helps to discover potential active compounds in complex matrices.

## Figures and Tables

**Figure 1 ijms-20-03875-f001:**
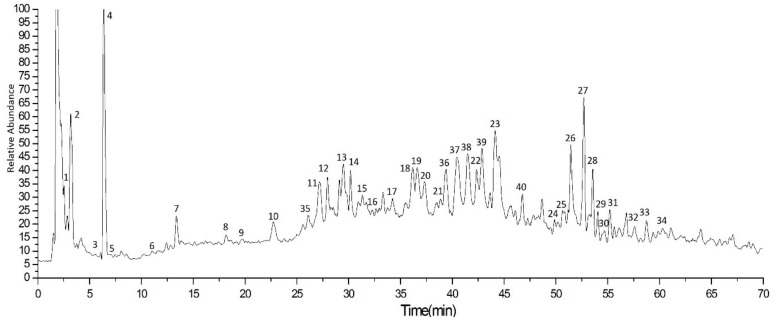
Total ion chromatogram of compounds in aqueous extracts from the *E. globulus* leaves using UPLC-Q-Orbitrap-MS.

**Figure 2 ijms-20-03875-f002:**
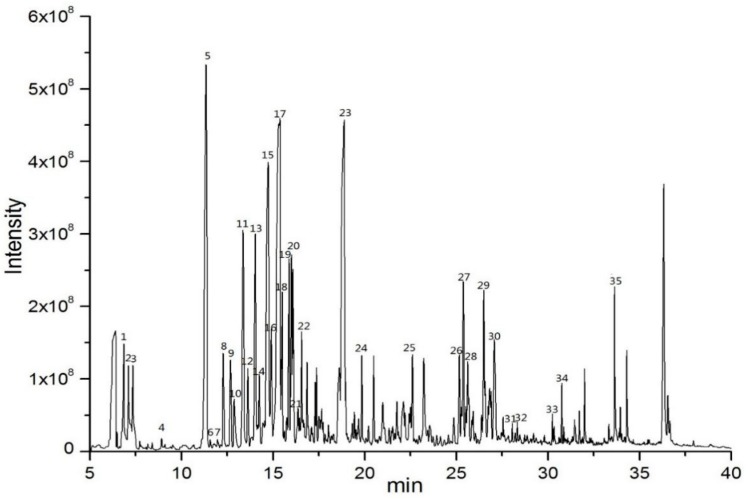
Total ion chromatogram of volatile compounds from the *E. globulus* leaves using HS-SPME-GC-MS.

**Figure 3 ijms-20-03875-f003:**
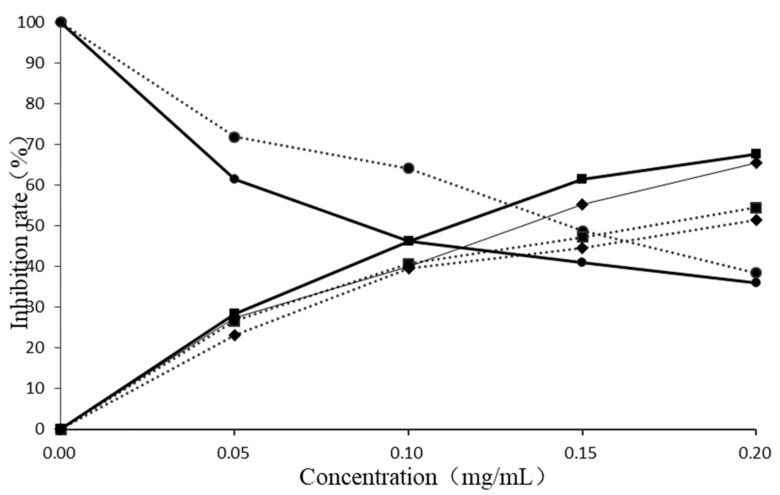
Allelopathic effects of *E. globulus* leaves extract on grass seeds. ● indicates the germination rate, ■ indicates the seedling inhibition rate, ♦ indicates the root inhibition rate, the solid line is the allelopathic effect of the aqueous extracts of the *E. globulus* leaves on grass seeds, and the dotted line indicates the allelopathic effect of the volatile components of *E. globulus* leaves on grass seeds.

**Figure 4 ijms-20-03875-f004:**
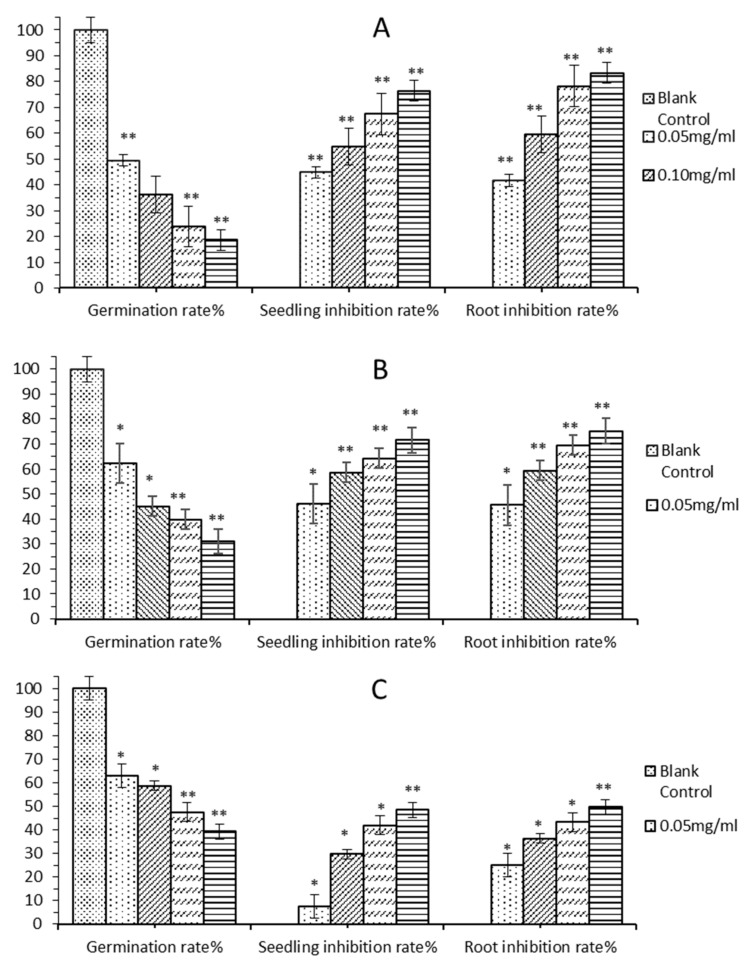
Allelopathic effects of predicted active compounds in *E. globulus* leaves **A** is gallic acid, **B** is citric acid, and **C** is isopulegol. * represents a significant difference between the experimental group and the control group (* *p* < 0.05, ** *p* < 0.01).

**Table 1 ijms-20-03875-t001:** Analysis of bioactive components of *E. globulus* leaves by UPLC-Q-Orbitrap-MS.

NO.	Identification	TR/min	Formula	[M-H]^−^	MS^2^	Relative Content (%)	Literature
1	Citric Acid	2.09	C_6_H_8_O_7_	191.0166	173.0061/111.0061/87.0063	0.19	[12]
2	Quinic acid	3.30	C_7_H_12_O_6_	191.0529	147.0270/129.0165/85.0271	2.40	[11]
3	Glucogallic acid	6.13	C_13_H_16_O_10_	331.0634	271.0425/169.0112	0.01	[11]
4	Gallic acid	6.31	C_7_H_6_O_5_	169.0111	125.0216	2.81	[11]
5	Ferulic acid	6.85	C_10_H_10_O_4_	193.0543	148.0589	0.01	[11]
6	Protocatechuic acid glucoside	12.96	C_13_H_16_O_9_	315.0685	153.0164	0.01	[11]
7	Protocatechuic acid	13.38	C_7_H_6_O_4_	153.0164	109.0269	0.15	[11]
8	Gentiopic glucoside	17.38	C_13_H_16_O_9_	315.0676	153.0162	0.01	[13]
9	2-O-caffeoylquinic acid	19.82	C_16_H_18_O_9_	353.0838	191.0529/179.0319/135.0424	0.02	[11]
10	2,5-dihydroxybenzoic acid	22.70	C_7_H_6_O_4_	153.0164	109.0269	0.16	[11]
11	Chlorogenic acid	26.93	C_16_H_18_O_8_	337.0894	191.0528/163.0369/119.0474	0.02	[14]
12	2-O-Coumaroylquinic Acid	27.99	C_16_H_18_O_8_	337.0894	163.0370/119.0475/119.0528	0.09	[13]
13	3,3′-Di-O-ellagic acid 4′-glucoside	29.61	C_34_H_24_O_22_	783.0594	765.0491/597.0453/300.9952	0.32	[14]
14	3-O-trans-caffeoylquinic acid	30.17	C_16_H_18_O_9_	353.0838	191.0531/179.0320	0.28	[15]
15	Caffeic acid	31.33	C_9_H_8_O_4_	179.0318	135.0422	0.15	[11]
16	cis-3-O-Coumaroylquinic Acid	32.31	C_16_H_18_O_9_	353.0838	191.0530/179.0320/173.0423/135.0421	0.01	[15]
17	2-O-trans-Coumaroylquinic Acid	34.63	C_16_H_18_O_8_	337.0894	173.0424/191.0527/163.0370	0.01	[13]
18	Apigenin-7-O-β-D-glucopyranoside	36.11	C_21_H_20_O_10_	431.0968	385.1826/205.1199153.0891	0.10	[15]
19	epigallocatechin	36.55	C_15_H_14_O_7_	305.0662	225.1100/96.9575	0.09	[15]
20	Apigenin-7-O-galactoside	37.40	C_21_H_20_O_10_	431.0969	367.0140/179.0532/89.0219	0.08	[16]
21	3-O-Coumaroylquinic Acid	38.50	C_16_H_18_O_8_	337.0894	191.0529/173.0424/93.0320	0.02	[15]
22	5,7-hydroxyl-2-(1-methylpropyl)isopropyl-chroone-8-β-D-galactoside	42.53	C_9_H_8_O_3_	163.0370	119.0475	0.07	[11]
23	4-O-Coumaroylquinic Acid	45.57	C_16_H_18_O_8_	337.0894	191.0530/161.0370	0.03	[13]
24	Ellagic acid-3-rhamnoside	50.84	C_20_H_16_O_12_	447.0547	300.9955	0.01	[15]
25	Benzoic acid	51.11	C_7_H_6_O_3_	137.0216	93.0321	0.04	[11]
26	Ellagic acid	51.52	C_14_H_6_O_8_	300.9953	300.9953	0.19	[14]
27	5,7-Hydroxyl-2-(1-methylpropyl)isopropyl-chroone-8-β-D-glucoside	52.61	C_19_H_24_O_9_	395.1311	275.0891/247.0947	0.03	[16]
28	Quercetin-3-O-glucuronide	52.70	C_21_H_18_O_13_	477.0616	301.0318	0.17	[17]
29	5,7-Hydroxyl-2-(1-methylpropyl)isopropyl-chroone-8-β-D-galactoside	54.02	C_19_H_24_O_9_	395.1286	275.0891/247.0934	0.03	[16]
30	Quercetin-3-O-arabinoside	54.68	C_20_H_18_O_11_	433.0710	301.0339	0.01	[18]
31	kaempferol-7-O-glucuronide	56.90	C_21_H_18_O_12_	461.0666	286.0371	0.07	[18]
32	Apigenin-7-O-glucuronide	58.42	C_21_H_18_O_11_	445.0767	269.0425/113.0217	0.09	[19]
33	3-O-methyl ellagic acid-O-glucopyranoside	58.83	C_21_H_18_O_12_	461.0675	315.0109	0.02	[20]
34	3,4-Dihydroxyhydrocinnamic acid	61.82	C_9_H_10_O_4_	181.0838	137.0943	0.01	[20]

**Table 2 ijms-20-03875-t002:** Analysis of volatile components of *E. globulus* leaves by HS-SPME-GC-MS.

NO.	Identification	TR(min)	Formula	RI	Relative Content (%)
1	a-Pinene	6.63	C_10_H_16_	933	9.8
2	Camphene	7.27	C_10_H_16_	950	2.43
3	β-pinene	7.39	C_10_H_16_	981	0.44
4	3-Carene	8.52	C_10_H_16_	995	0.80
5	α-Terpinolene	11.20	C_10_H_16_	1020	0.17
6	D-Limonene	11.51	C_10_H_16_	1032	2.59
7	Eucalyptol	11.99	C_10_H_18_O	1039	21.49
8	Ocimene	12.42	C_10_H_16_	1043	0.45
9	Gamma terpinene	12.51	C_10_H_16_	1061	0.52
10	Rosenoxide	12.63	C_10_H_18_O	1112	0.01
11	Fenchyl alcohol	12.83	C_10_H_18_O	1119	1.83
12	Camphor	13.77	C_10_H_16_O	1151	0.15
13	Isopulegol	13.88	C_10_H_18_O	1155	0.41
14	Borneol	14.39	C_10_H_18_O	1173	2.23
15	(-)-4-Terpineol	14.67	C_10_H_18_O	1182	0.86
16	(2R,5R)-2-Methyl-5-(prop-1-en-2-yl)cyclohexanone	14.99	C_10_H_16_O	1193	5.15
17	α-Terpineol	15.12	C_10_H_18_O	1198	0.22
18	Myrtenol	15.26	C_10_H_16_O	1202	0.43
19	Dihydrocarvone	15.51	C_10_H_16_O	1210	0.13
20	2-Oxabicyclo[2.2.2]octan-6-ol, 1,3,3-trimethyl-	15.99	C_10_H_18_O_2_	1224	0.31
21	Citronellol	16.59	C_10_H_20_O	1242	0.16
22	D-Carvone	16.66	C_10_H_14_O	1244	0.38
23	Menthone	17.03	C_10_H_16_O	1256	0.12
24	Citral	17.42	C_10_H_16_O	1267	0.20
25	(-)-β-elemene	23.18	C_15_H_24_	1395	0.16
26	β-Caryophyllene	25.08	C_15_H_24_	1430	2.42
27	Caryophyllene	25.42	C_15_H_24_	1436	0.14
28	α-Caryophyllene	26.81	C_15_H_24_	1462	0.37
29	(+)-aromadendrene	27.21	C_15_H_24_	1469	0.57
30	B-cadinene	27.75	C_15_H_24_	1479	0.17
31	β-selinene	28.42	C_15_H_24_	1491	0.13
32	1H-Cycloprop[e]azulene,1a,2,3,5,6,7,7a,7b-octahydro-1,1,4,7-tetramethyl-(1aR,7R,7aS,7bR)-	28.83	C15H24	1498	0.32
33	γ-cadinene	29.67	C15H24	1517	0.13
34	δ-cadinene	30.11	C15H24	1527	1.52
35	1H-Cycloprop[e]azulen-4-ol,decahydro-1,1,4,7-tetramethyl-,(1aR,4R,4aR,7R,7aS,7bS)-	32.69	C15H26O	1586	0.27

**Table 3 ijms-20-03875-t003:** Docking results of *E. globulus* leaves and 1BX9.

No.	Compound	Lig Score1	Lig Score2	PLP1	PLP2	Jain	PMF	Dock Score
1	Gallic acid	1.74	2.14	22.78	21.7	−0.43	57.46	89.142
2	Glucogallic acid	0.7	0.6	4.7	9.22	−1.91	29.48	82.628
3	Citric acid	2.07	1.95	14.4	20.6	−1.29	43.64	76.598
4	5,7-Hydroxyl-2-(1-methylpropyl)isopropyl-chroone-8-β-D-galactoside	2.38	2.93	31.52	30.75	0.46	107.74	72.267
5	3,4-DihydroxyhYdrocinnamic acid	1.93	2.29	23.41	38	0.41	65.15	66.73
6	3-O-trans-Coumaroylquinic acid	2.26	2.68	26.18	37.87	1.4	87.69	66.699
7	Caffeic acid	1.9	2.4	25.45	38.58	0.71	71.17	64.25
8	Quinic acid	1.9	1.58	11.05	14.78	1.13	43.47	62.489
9	5,7-Hydroxyl-2-(1-methylpropyl)isopropyl-chroone-8-β-D-glucoside	1.67	1.43	32.81	36.31	1.17	104.72	61.902
10	3-O-trans-Caffeoylquinic acid	2.51	2.78	30.51	42.8	1.29	96	56.973
11	cis-3-O-Coumaroylquinic acid	1.93	3.05	29.34	41.58	1.66	92.73	55.67
12	2-O-trans-Coumaroylquinic acid	3.23	2.94	26.54	38.98	0.64	89.46	53.726
13	Chlorogenic acid	2.17	2.8	25.04	41.94	1	101.15	52.818
14	2-O-Caffeoylquinic acid	2.47	3.11	38.91	45.61	0.59	94.34	52.714
15	Menthone	1.5	2.04	12.32	12.45	−1.09	46.31	52.166
16	Protocatechuic acid glucoside	2.21	3.3	29.87	46.23	0.76	67.21	49.487
17	2,5-Dihydroxybenzoic acid	1.56	2.47	13.35	14.67	−0.78	40.9	48.15
18	Benzoic acid	0.89	1.94	7.06	7.86	−1.07	40.42	47.551
19	Ferulic acid	1.74	3.07	17.6	28.85	0.38	59.74	47.087
20	Gentiopicroside	2.53	1.03	3.63	17.49	2.44	69.56	46.111
21	Protocatechuic acid	1.83	2.73	17.32	31.89	1.8	49.28	42.911
22	Epigallocatechin	2.1	3.23	39.58	51.66	1.8	88.34	40.993
23	Ellagic acid	2.65	3.3	34.44	36.64	-0.18	92.63	32.576
24	Isopulegol	2.3	3.01	30.07	34.74	1.51	59.87	25.029
25	Rose oxide	1.2	2.89	17.82	28.09	1.56	61.5	22.21
26	Menthone	1.5	2.93	15.7	16	0.61	46.76	21.555
27	Citral	1.51	2.97	22.2	30.08	0.84	58.13	21.089
28	2,5-dihydroxybenzoic acid	1.3	2.92	24.93	25.85	0.95	55.84	20.39
29	Fenchyl alcohol	1.16	2.7	23.37	22.92	0.14	56.24	19.142
30	Myrtenol	1.41	2.83	19.58	18.61	0.72	50.72	19.131
31	Dihydrocarvone	0.45	2.8	24.48	25.09	−0.13	60.37	19.017
32	(2R,5R)-2-Methyl-5-(prop-1-en-2-yl)cyclohexanone	0.43	2.78	25.81	25.34	−0.1	61.49	18.259
33	3-Carene	0.45	2.79	26.47	28.15	1.66	59.34	17.182
34	d-Limonene	0.38	2.66	24.21	25.95	−0.02	60.25	17.138
35	D-Carvone	0.51	2.89	27.49	27.13	−0.04	58.65	17.117
36	γ-Terpinene	0.42	2.73	25.09	26.37	0.71	61.9	17.116
37	Citronellol	1.27	2.85	25.08	28.96	−0.99	57.52	16.85
38	α-Terpineol	2.02	3.26	25.42	30.86	2.09	57.13	16.766
39	Camphene	0.35	2.61	24.46	23.88	0.71	57.78	15.827
40	Ocimene	0.58	3.01	24.62	28.25	0.4	61.88	14.504
41	Terpinyl acetate	0.93	2.87	26.47	25.1	−0.4	63.72	14.124
42	β-Elemene	0.53	2.91	26.79	25.52	−0.82	68.55	13.708
43	β-Eudesmol	1	2.83	29.84	35.44	0.68	74.6	12.253
44	Globulol	0.37	2.68	16.5	18.05	−1.69	18.5	11.926
45	Borneol	1.19	2.69	13.59	15.4	−0.07	46.34	11.754
46	α-Terpinolene	0.16	2.28	17.95	28.24	1.39	62.53	11.285
47	β-Pinene	0.19	2.32	25.08	26.6	0.51	56.72	10.315
48	Cineole	0.18	1.6	24.83	27.37	1.77	58.45	9.034
49	α-Pinene	0.09	2.17	24.55	25.87	0.77	59.29	7.723
50	Camphanone	0.52	1.89	23.33	26.3	0.76	58.95	4.111
51	γ-Cadinene	0.31	2.5	24.9	28.05	2.53	75.53	4.041
52	β-Cadinene	-0.24	1.62	23.67	37.92	0.96	60.27	2.74

**Table 4 ijms-20-03875-t004:** Docking model of active compounds.

Compound	Absolute Energy	Binding Mode	H-Bind Number	Residues Involved in H-Bond Formation
Gallic acid	15.277	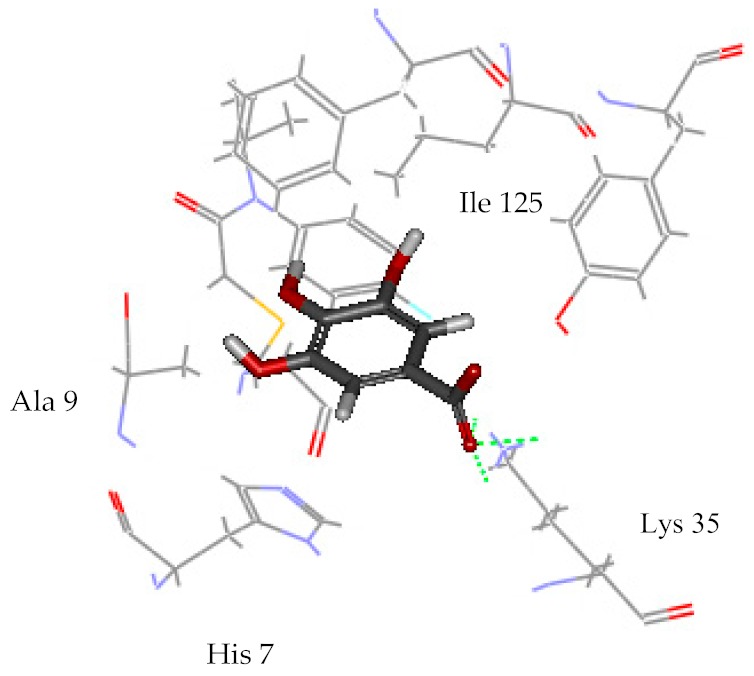	3	Lys 35
Glucogallic acid	13.502	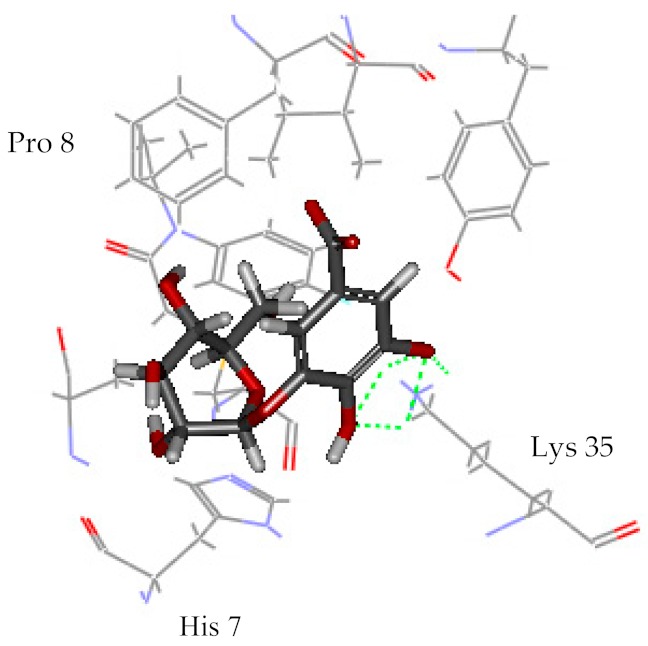	5	Lys 35
Citric acid	12.893	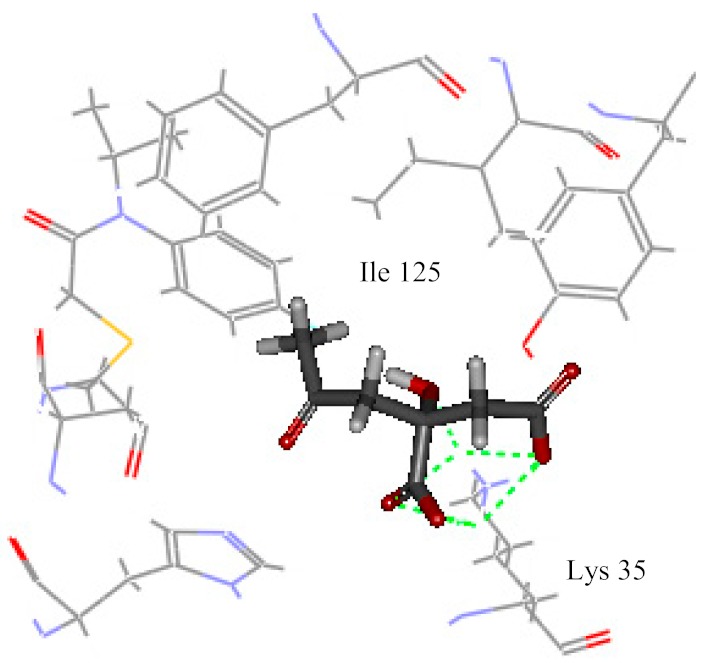	5	Lys 35
Isopulegol	4.431	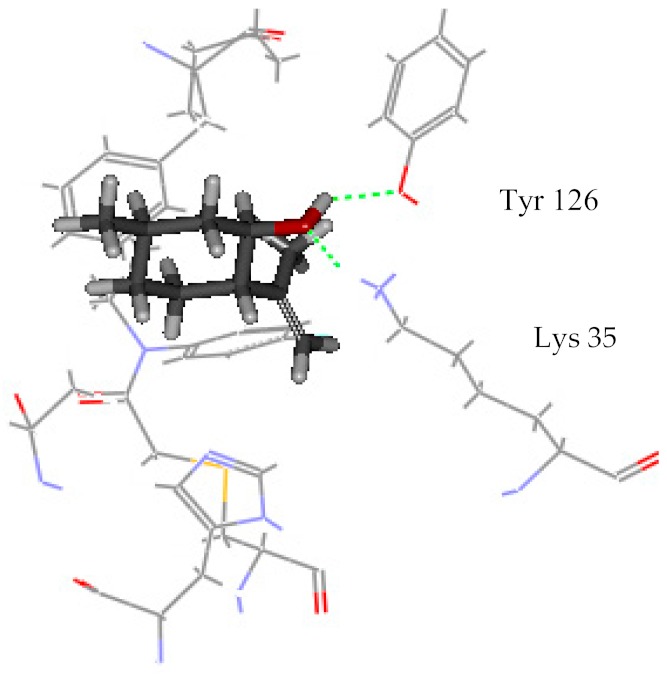	2	Lys 35 Tyr 126

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
