# Peer review of "A Strategy Based on GC-MS/MS, UPLC-MS/MS and Virtual Molecular Docking for Analysis and Prediction of Bioactive Compounds in Eucalyptus Globulus Leaves"

_ijms, 2019, doi:10.3390/ijms20163875_

Round 1
Reviewer 1 Report
This is an interesting manuscript that presents research on the use of analytical techniques and virtual molecular docking for prediction of bioactive compounds. It is obvious that significant work went into the research described and the results are very interesting. However, some aspects of the paper should be clarified or corrected:
- Scientific names (i.e. Eucaliptus globulus) must be in italics throughout the manuscript. First time the complete name, later only species name (E. globulus).
- Figure 4A: legend is incomplete.
- 3.2.1 and 3.2.2: you say “fresh Eucalyptus”. I suppose that are the air dried leaves. Fresh seems to indicate not dried leaves. Please clarify.
- 3.2.1 and 3.2.2: space between 10 or 100 and g (i.e. 10 g, 100 g).
- P.14, l. 280: change 50/30 mm to 50/30 µm.
- P. 10, l. 335: provide a reference for the test method GB2772-81.
- Provide the company name for the software you have used (i.e. p16, l339: Microsoft Excel 201*)
Author Response
Reviewer #1
This is an interesting manuscript that presents research on the use of analytical techniques and virtual molecular docking for prediction of bioactive compounds. It is obvious that significant work went into the research described and the results are very interesting. However, some aspects of the paper should be clarified or corrected:
Scientific names (i.e. Eucaliptus globulus) must be in italics throughout the manuscript. First time the complete name, later only species name (E. globulus).
Response 1: Thank you for pointing this out. Eucalyptus globulus has been revised.
Figure 4A: legend is incomplete.
Response : The illustration of figure 4A has been modified.
2.1 and 3.2.2: you say “fresh Eucalyptus”. I suppose that are the air dried leaves. Fresh seems to indicate not dried leaves. Please clarify.
Response: we did use fresh E. globulus leaves in our study. 3.1 about E. globulus leaves has been corrected.
2.1 and 3.2.2: space between 10 or 100 and g (i.e. 10 g, 100 g).
Response : It has been revised
p.14, l. 280: change 50/30 mm to 50/30 µm.
Response: 50/30 mm has been revised to 50/30 µm.
p. 10, l. 335: provide a reference for the test method GB2772-81.
Response: The test method GB2772-81 has been revised to GB2772-99, and has been provided as supplementary material Supplement1.
Provide the company name for the software you have used (i.e. p16, l339: Microsoft Excel 201*)
Response: The software name has been provided.
Reviewer 2 Report
The manuscript titled “A strategy based on GC-MS/MS, UPLC-MS/MS and virtual molecular docking for analysis and prediction of bioactive compounds in Eucalyptus globulus leaves” reports a combined methodology using GC–MS/MS, UPLC-MS/MS and molecular docking to identify bioactive compounds in natural products. The manuscript can be accepted after a minor revision.
The structures for all the compounds should be provided.Author Response
Reviewer # 2
The manuscript titled“A strategy based on GC-MS/MS, UPLC-MS/MS and virtual molecular docking for analysis and prediction of bioactive compounds in Eucalyptus globulus leaves”reports a combined methodology using GC-MS/MS, UPLC-MS/MS and molecular docking to identify bioactive compounds in natural products. The manuscript can be accepted after a minor revision.
The structures for all the compounds should be provided.
Response: All the structures has been provided as supplementary material Supplement 2.